# Family correlates of emotional and behavioral problems in Nepali school children

Jasmine Ma[1,2]*, Pashupati Mahat[3], Per Håkan Brøndbo[4], Bjørn H. Handegård[1], Siv Kvernmo[5], Anne Cecilie Javo[1,6]

1 Regional Centre for Child and Youth Mental Health and Child Welfare -North, Faculty of Health Sciences, UiT The Arctic University of Norway, Tromsø, Norway, 2 Child & Adolescent Psychiatry Clinic, Kanti Children's Hospital, Kathmandu, Nepal, 3 Centre for Mental Health and Counseling, Kathmandu, Nepal, 4 Department of Psychology, Faculty of Health Sciences, UiT The Arctic University of Norway, Tromsø, Norway, 5 Department of Clinical Medicine, Faculty of Health Sciences, UiT The Arctic University of Norway, Tromsø, Norway, 6 Sami National Competence Center for Mental Health, Sami Klinihkka, Finnmark Hospital Trust, Karasjok, Norway

* jasminema2006@yahoo.com

## Abstract

### Background

There is a substantial gap in our knowledge about family correlates of child emotional and behavioral problems in low- and middle-income countries (LMIC). The present study contributes to filling this gap by examining such correlates in a larger population study in Nepal.

### Methods

Our study is a cross-sectional, observational study among 3840 Nepali children aged 6–18 years from 64 schools and 16 districts in the three main geographical regions in the country. We used the Nepali version of the Child Behavior Checklist (CBCL)/6-18 to assess children's internalizing and externalizing problems and an additional background information questionnaire to assess possible family correlates which included parental education, family structure, migrant worker parents, parental mental and physical illness, family conflicts, and child-rearing. The associations between family variables and child internalizing and externalizing problems were analyzed using bivariate correlations and multiple regression.

### Results

Using bivariate analysis, we found that mental and physical illness in parents, conflict in the family, parental disagreement in child-rearing, and physical punishment of child correlated positively with both Internalizing Problems and Externalizing Problems. The same associations were found by using multiple regression analysis. In addition, parental education, family structure, and migrant worker mothers were associated with Externalizing Problems. However, the effect sizes were small.

**Data Availability Statement:** All relevant data are within the manuscript and Supporting information files.

**Funding:** This study is funded by Child Workers in Nepal (CWIN) / Solidarity Action for Development,

Norway FORUT. The funders had no role in the
study design, data collection and analysis, decision
to publish, or preparation of the manuscript.

**Competing interests:** The authors have declared
that no competing interests exist.

## Conclusion

The results suggest that in Nepal, child mental problems were associated with several family
risk factors. Further, the study points to the need of strengthening prevention- and interven-
tion measures to minimize family risk factors of child mental health disorders.

## Introduction

Children constitute 42% of the world's population and roughly 90% of these live in low- and
middle-income countries (LMIC) [1]. Children in LMIC have increased risk for mental illness
due to social and environmental conditions such as poverty and lack of child mental health
services [2, 3]. There is a substantial gap in economic and human resources for child mental
health services, particularly in LMICs. Documenting children's mental health problems and
associated risk factors can provide stronger arguments for care. The present study aims at
providing new epidemiological knowledge about family correlates of child emotional and
behavioral problems (EBP) in the general child population of Nepal. In a previous, recently
published paper, we reported data from the same sample on the prevalence, magnitude and
types of child EBP in different castes and ethnic groups in Nepal and in different geographical
locations and types of living area. The level of problems was found to be rather high, demand-
ing additional investigations into possible family risk factors [4].

Not much research has been done on the association between environmental risk factors
and child behavior problems in LMICs, including Nepal. Till now, there is no documentation
on family correlates of child EBP in Nepal on a national level, and very few studies on child
mental health have been conducted altogether in the country. Searching the data bases
(PubMed, Google scholar, and PsycINFO), only two small-scale Nepali studies on correlates of
child EBP were found. One study, which was done among adolescents in Hetauda Municipal-
ity in Central Nepal, found that adolescents whose families had frequent disputes, adolescents
from nuclear families or adolescents living with a single parent, and adolescents of illiterate
parents were more likely to have a psychosocial dysfunction [5]. A qualitative study on chil-
dren done in a rural area in Chitwan district, Tarai region, suggested that an unfavorable fam-
ily environment and physical punishment might lead to increase of emotional problems in
children [6]. This sparse amount of research points to the fact that more and larger epidemio-
logical studies are warranted, especially studies on a national level.

The international literature has consistently documented the influence of environment on
children's behavior and psychosocial functioning [7, 8]. Worldwide, several types of environ-
mental risk factors have been found to associate with EBP, including specific family- and par-
enting factors [9, 10].

As for the effect of parental education level on child EBP, studies have found that a lower
parental education level associates with more child behavioral problems and less psychological
wellbeing [11–13]. Children from families with higher educated parents showed lower risk of
mental health problems than their peers with less educated parents [14, 15].

Further, literature suggests that children raised by single mothers may be at increased risk
of child EBP [16, 17]. In studies examining extended family households as a potential risk fac-
tor of child EBP, results are ambiguous. Some studies have suggested that living in an extended
family has a positive impact on children [18], whereas other studies show higher levels of EBP
in extended families than in nuclear families [19–21]. The cultural and economic context of
families may probably explain the discrepancy. In some societies, living in an extended family

may be a sign of low socioeconomic status and lack of resources, whereas in other societies it may be culturally established as a good way of organizing family life, offering several advantages. However, few studies have examined the effect of family structure (i.e. single motherhood, versus nuclear family, versus extended family) on child EBP in LMICs, including Nepal.

Another factor pertaining to family life that may influence child EBP, is the increasing number of migrant worker parents in search of employment opportunities, leaving their children back home. Migrant worker parents are a common phenomenon in many LMICs. A recent systematic review and meta-analysis study showed that compared with children of non-migrant worker parents, left-behind children had increased risk of mental health problems [22]. Several studies from China reported that left-behind children experienced more mental health problems, poorer school performance, and early school dropouts [23–25].

A large body of research has demonstrated that a stressful family life caused by parental psychopathology, somatic illness in parents, family conflicts, as well as impaired parenting may affect children's mental health and psychosocial development [26, 27]. For instance, it has been found that parental psychopathology may increase the risk for both internalizing and externalizing behavior problems in children [27–29]. Similarly, studies suggest that parents' physical illness can lead to negative psychological outcome in children, including increased rates of internalizing and externalizing behavior problems [30, 31]. Further, conflicts in the family, in particular marital conflict and disagreement about child-rearing, emerge as significant risk factors for the development of psychopathology in children, either directly or indirectly [32–34]. Parental disagreement in child-rearing is found to associate with both internalizing and externalizing problems in children [35, 36].

As for harsh parenting, such as physical punishment, studies from many societies show a positive association with child EBP. A comprehensive meta-analysis including 160,927 children from both US based- and international studies found that spanking associated with both internalizing and externalizing child behaviors [37]. Another meta-analysis with children from several countries found that physical punishment was associated with adverse child outcomes, especially in countries in which physical punishment was less culturally normative [38]. In Nepal, physical punishment of children is widely accepted [39]. A recent Nepali study suggested that a series of negative behaviors in children may provoke physical punishment by parents as well as by teachers [6]. However, both in Nepal and in other LMICs, few studies have explored the effect of physical punishment on children's behavior on a larger scale.

In the present study, which is part of a large-scale epidemiological project [4], our specific aim was to assess the associations between selected family variables and internalizing and externalizing behavior problems in Nepali schoolchildren aged 6–18 years.

## Materials and methods

### Study design

The present study is a cross-sectional, observational study in the general population of Nepal.

### Study site and population

Nepal is a mountainous country and is topographically divided into three regions: the Himalaya (Mountain region) to the north, the Middle Hills region, which lies between the northern- and southern belts, and the Terai region to the south. The capital city Kathmandu lies in the Middle Hills region. Nepal has a population of 29.1 million people (2020). Nepal is ranked among the low- and middle-income countries (LMIC) with per capita nominal Gross Domestic Product (GDP) of 1,090 US$ [40]. According to the Nepal Demographic and Health Survey (2016), about 30% of all women and 10% of all men were illiterate, and 17% of women and

19% of men had attended primary school only [41]. According to the Central Bureau of Statistic (CBS) Nepal, 2011, children below 18 years of age represent 44.4% of the total population: 22.5% boys and 21.9% girls. Female-headed households (single motherhood) were 25.7% of all households. Most single mothers were widows [42].

## Subjects and procedure

This study was conducted in the 3 main geographical regions of Nepal where we purposively selected 3 districts from the Mountain region, 6 districts each from the Middle Hills and the Tarai regions, which added up to 15 districts. In addition, the Kathmandu district was included which makes it 16 districts in total.

Further, we purposively selected 4 schools (two government and two private schools) in each district based on accessibility and convenience, i.e., 64 schools in 16 districts. 6 students equally distributed across gender were randomly selected in each grade, from grade 1–10. Thus, in each district, 240 children were selected, which gave a total of 3840 children. The overall participation rate was 99.5% (i.e., 3820 students).

**Procedure.** After an approval was received from the Ministry of Education, Nepal, meetings were scheduled with the school coordinators and school principals and information about the study and the reasons for carrying out the research were provided. Data collection was performed by 30 trained research assistants (RA) and 7 trained field supervisors who were in turn supervised by the researcher. Once the students were randomly selected, their mothers were invited to the school with the help of the school administrator who sent an invitation letter to the families. Both oral and written information about the study were provided. Informed consent was then obtained from the parents by the RAs and confidentiality was assured. Those parents not showing up were informed by home visits by the RAs and invited to participate. For illiterate parents, the RAs verbally posed the questions to them, and helped fill in the forms. The parents were encouraged to respond to all items in the questionnaires and were given the opportunity to ask questions about any item. A small gift was given as an acknowledgement of their participation in the study. The data were collected during September 2017– January 2018, and the plotting in of data was done manually during the first half of 2018 by three research assistants, monitored and supervised by the researcher. The proportion of missing items was not more than 0.1% for any of the items included in the instruments.

## Measures

**1. Child Behavior Checklist (CBCL 6–18).** For this study, we used the Nepali version of the Child Behavior Checklist for ages 6–18 (CBCL/6-18) that had been translated into Nepali language in connection with a former Nepali study [43]. The CBCL/6-18 consists of 20 competence items and 120 problem items. The problem items are scored on 8 syndrome scales, 2 broadband scales: Internalizing and Externalizing, and a Total Problems scale. The syndrome scales: Withdrawn/Depressed, Somatic Complaints and Anxious/Depressed together form the "Internalizing" scale, and the scales: Rule-breaking behavior and Aggressive behavior together form the "Externalizing" scale. The Social Problems, Attention Problems, and Thought Problems scales do not belong to either of the broadband scales, but are included in the Total Problems scale, which is derived by summing up the individual item scores. The response format of questions on behaviors is: 0 = not true, 1 = somewhat or sometimes true, and 2 = very true or often true.

CBCL has been translated into more than 100 languages and has established good psychometric properties cross-culturally [44]. It has been found to have strong validity and good test-retest reliability and internal consistency. For empirically based syndrome- and competence

scales, the mean test-retest reliability was 0.90. Internal consistencies of the syndrome scales measured by Cronbach's alpha ranged from 0.78 to 0.94 [44].

To assess the internal consistency of the Nepali version of CBCL/6-18 for the present study, we computed Cronbach's alphas for the empirically based syndrome scales. The results are reported in a previous paper [4] and were: Withdrawn / Depressed: 0.71; Somatic Complaints: 0.79; Anxious / Depressed: 0.76; Rule-breaking Behavior: 0.76; Aggressive Behavior: 0.88; Social Problems: 0.73; Attention Problems: 0.80; Thought Problems: 0.75 [4].

**2. Background information questionnaire.** The parents were asked to fill in a questionnaire asking about background information. Selected family variables for this study were: 1) Parental education level which was categorized as follows: (a) no education (illiterate); (b) 1–8 years of education; (c) 9–12 year of education; and (d) more than 12 years of education. (2) Family structure, questions being asked with three options: (a) child living with a single parent, (b) nuclear family, or (c) extended family. "Single parenthood" was defined as living with a single parent (either a widow, a divorced, or separated parent). "Nuclear family" was defined as living with both parents and siblings, and "Extended family" as living with parents, siblings, grandparents and/or immediate relatives. (3) Migrant worker parents, the question being asked with a "yes" or a "no" option.

Questions about family life and child-rearing included: (4) Parental illness, the questions posed being whether any of the parents had any mental illness or had any physical illness or disabilities. The questions were asked with a "yes" or a "no" option and the parent was encouraged to explain further about symptoms if the answer was a "yes". Further, parents were asked about (5) Conflicts within the family, the question posed being: "Has there been any conflicts between family members causing stress in the family during the past 6 months?" The options offered were high, moderate, or low level of conflict. (6) Agreement in child-rearing was asked by posing the question: "Do you as parents agree as to child-rearing?", the options being "highly agree", "somewhat agree", and "totally disagree". As for (7) child-rearing methods, we asked whether the parents made frequently use of physical punishment to control the child's misbehavior. The question was asked with a "yes" or a "no" option.

## Statistical analyses

The ASEBA data management and SPSS statistics version 26.0 for Windows were used for all analyses. First, bivariate correlations (Pearson correlation and Kendall's tau-b) were examined to assess the association between child internalizing or externalizing behaviors and family variables. Then, multiple regression analyses were done to assess the associations between the different independent variables and child behavior problems. In these regression analyses, all the independent variables entered the model. Child age, child gender, and traumatic life events were used as control variables. Main effects of the different correlates were then tested. Partial eta squared was selected for measuring the effect size. The significance level used for all tests was 0.005.

## Ethical considerations and confidentiality of data

Ethical approval was obtained from the Ethical Review Board of Nepal Health Research Council (NHRC) (ref. no. 1875; reg, no: 71/2017). Both collection and storage of data were done according to their rules. The records from the study were kept strictly confidential and locked down so that no persons other than the researcher had access to them. All electronic information is coded and secured using a password protected file, and all the personally identifiable information has been removed from the dataset to protect the participants'

individual privacy. No information will be shared or published that would make it possible to identify any participant.

## Results

### Background variables

The background variables of the sample are presented in Table 1. As seen in the table, about 11% of the parents were illiterate. The majority (44%) had a primary level education only. Almost 20% of all parents were migrant workers, mostly fathers. Half of the children lived in a nuclear family. Almost a quarter of all parents had some kind of physical illness, but few parents reported a mental illness (3%). About 12% of the parents reported a moderate to high level of conflict in the family and about 36% reported a moderate to high disagreement in child-rearing. Few parents reported a frequent use of physical punishment of the child to control misbehavior (10%).

**Table 1. Background table.** Distribution of family variables.

| Background variables | Total sample (N = 3820) |
|---|---|
| **Parental education level** | |
| Illiterate | 417 (10.9%) |
| Primary level (grade 1 to 8) | 1694 (44.3%) |
| Secondary level (grade 9–12) | 1309 (34.3%) |
| University level (Bachelor, Masters, PhD) | 400 (10.5%) |
| **Family structure** | |
| Single parent | 380 (9.9%) |
| Nuclear family | 2093 (54.8%) |
| Extended family | 1347 (35.3%) |
| **Migrant worker mother** | |
| Yes | 66 (1.7%) |
| No | 3754 (98.3%) |
| **Migrant worker father** | |
| Yes | 665 (17.4%) |
| No | 3119 (81.6%) |
| **Mental illness in parents** | |
| Yes | 120 (3.1%) |
| No | 3700 (96.9%) |
| **Physical illness in parents** | |
| Yes | 910 (23.8%) |
| No | 2910 (76.2%) |
| **Conflict in the family causing stress in the past 6 months** | |
| Low level of conflict | 3351 (87.8%) |
| Moderate level of conflict | 383 (10.0%) |
| High level of conflict | 86 (2.3%) |
| **Parental disagreement in child-rearing** | |
| Little disagreement | 2451 (64.2%) |
| Somewhat disagrees | 1085 (28.4%) |
| Highly disagrees | 284 (7.4%) |
| **Use of physical punishment to control the child** | |
| Yes | 384 (10.1%) |
| No | 3436 (89.9%) |

**Table 2. Bivariate correlations between family variables and child internalizing and externalizing problems.**

| Family variables | Internalizing Problems | Externalizing Problems |
|---|---|---|
| Parents' years of education | 0.009 [a] | 0.020 [a] |
| Family structure | 0.006 [a] | 0.001 [a] |
| Migrant worker mother | 0.010 [b] | 0.039 [b] |
| Migrant worker father | 0.019 [b] | 0.025 [b] |
| Parental mental illness | 0.100 [b] * | 0.091 [b] * |
| Parental physical illness | 0.181 [b] * | 0.134 [b] * |
| Conflict in the family causing stress | 0.081 [a] * | 0.080 [a] * |
| Parental disagreement in child-rearing | 0.059 [a] * | 0.083 [a] * |

$p < 0.005$ = *;

[a] = Kendall's tau-b;

[b] = Pearson correlation.

**Family correlates of child EBP.** When examining correlations between family variables and child EBP, we used both descriptive, bivariate, first-order correlation analyses and multiple, linear regression analyses.

*Bivariate correlations.* Table 2 shows the bivariate correlations. Using a significance level of 0.005, we found that mental and physical illness in parents, conflicts in the family, parental disagreement in child-rearing, and physical punishment of child correlated positively with both internalizing and externalizing problems.

*Multiple regression analyses.* Multiple regression analyses were performed to examine the association between family variables and child Internalizing Problems and Externalizing Problems (Tables 3 and 4). For the Internalizing case, when adding the family correlates to a model already containing the control variables (child age, child gender, and life events), the proportion of the total variance explained increased from $R^2 = 0.037$ to $R^2 = 0.087$, and for the externalizing, $R^2$ increased from 0.027 to 0.083. When all the family variables and control variables were entered simultaneously, parents' education level, mental and physical illness in parents, and parental disagreement in child-rearing were significantly correlated with child Internalizing problems, but not family structure, migrant worker mother, migrant worker father, and physical punishment of child. For Externalizing problems, all family variables except for migrant worker father were significant.

Children of parents with 9 to 12 years of education scored the highest and children of parents with no education the lowest. Children from extended families scored higher, whereas children from single parent families had the lowest scores. Children whose mothers were migrant workers scored higher than children whose mothers were not migrant workers. Children with parents who had a mental illness scored higher than children whose parents were not mentally ill. Children with physically ill parents scored higher than children whose parents were not physically ill. Children belonging to families that experienced high level of conflict in the past 6 months scored higher than children belonging to families with a low level of conflict. Children experiencing frequent physical punishment scored higher than those not experiencing physical punishment.

## Discussion

In this study, we found significant associations between child EBP and family variables, such as: parental level of education, family structure, mental- and physical illness in parents, family conflicts, and child-rearing practices. However, the variables included in the study only

**Table 3. Multiple regression analysis of associations between family factors and internalizing problems.**

| Variables | F | B | SE | Partial eta squared |
|---|---|---|---|---|
| **Parental Education level** (Reference group: Illiterate-0 years of education) | 6.774 | | | 0.005** |
| Primary School (1–8 grade) | | 1.698 | 0.430 | 0.004** |
| Secondary School (9–12 grade) | | 1.783 | 0.448 | 0.004** |
| University education (Bachelor, Masters, PhD) | | 0.864 | 0.553 | 0.001 |
| **Family Structure** (Reference group: Nuclear family) | 3.823 | | | 0.002 |
| Single family | | -0.219 | 0.438 | 0.000 |
| Extended family | | 0.683 | 0.271 | 0.002 |
| **Migrant worker mother** (Reference group: Non-migrant worker mother) | 1.570 | 1.218 | 0.972 | 0.000 |
| **Migrant worker father** (Reference group: Non-migrant worker father) | 0.159 | -0.133 | 0.333 | 0.000 |
| **Parental mental illness** (Reference group: No mental illness) | 11.305 | 2.459 | 0.731 | 0.003* |
| **Parental physical illness** (Reference group: No physical illness) | 83.475 | 2.749 | 0.301 | 0.022** |
| **Family conflict** (Reference group: Low level of conflict) | 18.883 | | | 0.010** |
| High level of conflict | | 2.198 | 0.866 | 0.002 |
| Moderate level of conflict | | 2.432 | 0.425 | 0.009** |
| **Parental disagreement in child-rearing** (Reference group: Low level of parental disagreement) | 5.819 | | | 0.003* |
| Highly disagree | | 0.886 | 0.499 | 0.001 |
| Somewhat disagree | | 0.929 | 0.290 | 0.003* |
| **Use of physical punishment of child** (Reference group: No frequent use of physical punishment of child) | 5.892 | 1.034 | 0.426 | 0.002 |
| R² (control variables) | 0.037 | | | |
| R² (full model) | 0.088 | | | |

*p < .005;

**p < .0005.

F = "F-test statistic"; B = "unstandardized regression coefficient"; SE = "Standard error".

explained about 8–9% of the total variance in internalizing and externalizing problems, and the effect sizes were small for all included family variables.

When examining the association between parental education and child EBP, we found that children with parents who had 9–12 years of education tended to have more EBP, whereas children of illiterate parents had less problems. This result contrasts with other studies where children of the lowest educated parents had more EBP [11–13]. One explanation may be the inclusion of illiterate parents from indigenous groups/ ethnic minorities in our sample [4]. International studies have reported that parents of ethnic minorities maybe less likely to perceive EBP in their children as compared to parents of ethnic majority groups due to less acknowledgement of such problems [45]. Further, their reports of a lower amount of EBP may be because some of them had language difficulties as their home language was other than Nepali. Hence, they might have had difficulties in understanding the meaning of some of the questions and therefore chose the option with the lowest score.

We found that children living in extended families had more externalizing problems. Statistically, it was a larger difference between extended and single families than between extended and nuclear families. Our finding is consistent with a recent study from the USA which reported that children living in extended family households had more EBP than children living in nuclear family households [19]. Most of the present research in Western countries on family structure focuses on the presence or absence of a child's biological parents in a household, and on parents' marital or cohabitation status [46, 47]. However, an exclusive focus on nuclear family organization might produce an incomplete account of how family structure is related to child behavior and development. In Nepal, as in most LMICs, children often live-in

**Table 4. Multiple regression analysis of associations between family factors and externalizing problems.**

| Variable | F | B | SE | Partial eta squared |
|---|---|---|---|---|
| **Family education** (Reference group: Illiterate i.e. 0 education) | 5.040 | | | 0.004* |
| Primary school (1–8 grade) | | 1.560 | 0.429 | 0.003** |
| Secondary school (9–12 grade) | | 1.658 | 0.447 | 0.004** |
| University education (Bachelors, Masters, PhD) | | 1.365 | 0.552 | 0.002 |
| **Family Structure** (Reference group: Extended family) | 7.010 | | | 0.004* |
| Single family | | -1.601 | 0.455 | 0.003** |
| Nuclear family | | -0.670 | 0.270 | 0.002 |
| **Migrant worker mother** (Reference group: Non-migrant worker mother) | 9.651 | 3.013 | 0.970 | 0.003* |
| **Migrant worker father** (Reference group: Non-migrant worker father) | 1.305 | -0.380 | 0.333 | 0.000 |
| **Parents' mental illness** (Reference group: No mental illness) | 11.913 | 2.519 | 0.730 | 0.003* |
| **Parents' physical illness** (Reference group: No physical illness) | 47.468 | 2.069 | 0.300 | 0.012** |
| **Family conflict** (Reference group: Low level of conflict) | 17.766 | | | 0.009** |
| High level of conflict | | 2.671 | 0.864 | 0.003* |
| Moderate level of conflict | | 2.228 | 0.424 | 0.007** |
| **Parental disagreement in child-rearing** (Reference group: Low level of parental disagreement) | 9.736 | | | 0.005** |
| Highly disagree | | 1.009 | 0.498 | 0.001 |
| Somewhat disagree | | 1.227 | 0.289 | 0.005** |
| **Use of physical punishment of child** (Reference group: No frequent use of physical punishment of child) | 37.246 | 2.593 | 0.425 | 0.010** |
| $R^2$ (control variables) | | | 0.027 | |
| $R^2$ (full model) | | | 0.083 | |

*$p < .005$;

**$p < .0005$.

F = "F-test statistic"; B = "unstandardized regression coefficient"; SE = "Standard error".

multigenerational households with grandparents, aunts, uncles, and other relatives such as cousins, collectively referred to as 'extended family'. Internationally, research has shown that living in extended family households may be disadvantageous for child behavioral development, and mostly among married parent extended family households [48]. Others have found that the influence of extended household structures on children's cognitive and behavior development and emotional regulation might differ in different ethnic groups [20, 21]. Previous studies have found that extended family members may increase family stress by taking up housing space and through conflicts and negative interactions with parents, or they may change the distribution of resources within the household by absorbing resources that would otherwise be used for the children [49]. Extended families might include elderly members requiring care and attention by parents which may lead to decreased interactions between parents and children affecting child behavioral adjustment and psychological health [48]. However, empirical evidence to support this idea is lacking. On the other hand, extended family households may include extra caregivers who also provide extra income which may be beneficial to children's behavior development [18]. Further, studies have found that children who grow up with their grandparents live longer and are happier, and that grandparents may play an important part in solving family conflicts and in children's emotion regulation through their compassion and wisdom [50]. In Nepal, few studies on the association between family structure and child EBP have been published. Hopefully, our study may serve as a springboard for future, more detailed studies in this field of research.

In the present study, we found a positive association between parental mental illness and child behavioral problems. This result is consistent with previous findings that parental mental

illness negatively influences child EBP [27–29]. Parent's mental illness is clearly a risk factor for psychiatric disturbances in offspring, operating through a variety of genetic, psychological, and interactive mechanisms. Studies have demonstrated that there is a strong genetic link between parent and child psychology [51]. However, an examination of genetic links laid outside the scope of the present study- the focus of which was on the socio-environmental correlates of child EBP. Other studies have found that mental illness in parents can lead to relationship discord between parent and child, poor general parenting skills, social isolation of family, and poverty which can place children at much greater risk of impaired social, psychological and physical health than other children [52]. In addition, there may be a reciprocal relationship between parent's behaviors and child behaviors. It is conceivable that children who consistently observe internalizing behaviors (e.g., sitting alone, crying) or externalizing behaviors (e.g., yelling, shouting, hitting) in adults are more likely to imitate these behaviors than children who have never or rarely observed these kinds of behaviors [53, 54]. In our study, we found a low percentage of mental illness in parents. In a LMIC like Nepal, stigma and lack of awareness related to mental illnesses might be the reason for this low percentage. The real numbers of mental illnesses might be much higher. When parents do not seek treatment for their mental illness, self-blame on part of the child may result and this may accentuate other risk factors [55].

Similarly, our study showed a significant association between physical illness in parents and child internalizing and externalizing problems. This finding is in accordance with previous studies [31, 56]. One possible explanation may be the disruption of parenting due to lack of energy and stamina, or inability to provide adequate attention to their children due to their physical health issues [57]. Also, parental poor health might influence parental perception of their child's health. Studies have shown that mothers who self-reported poor health had increased odds of reporting their children with poor health [58]. However, the relationship between parents' illness and child EBP is not fully understood. Different illnesses are believed to affect families in different ways, and different kinds of adaptive behaviors may be required. A more thorough investigation into these matters is warranted.

We found a significant association between conflicts in the family causing a high level of stress, and child EBP. Our finding is in line with other international studies which show that conflicts in the family, particularly between parents, is a significant risk factor for the development of psychopathology in children [59, 60]. Generally, the effect of family conflict on children is explained by the extent to which one or more aspects of emotional security are negatively affected and the extent to which children are able to regulate general emotional distress [32, 34]. There are several potential mechanisms through which conflicts in a family might influence the occurrence of child EBP. For example, children's emotional security can be enhanced or undermined by the quality of the parent-child relationship, the context of a conflict, and conflicts between parents. However, exploring such mechanisms was beyond the scope of the present study.

Our study found a positive correlation between parental disagreement in child-rearing and both internalizing and externalizing problems. Our findings replicate the results from previous international studies. A meta-analysis by Teubert and Pinquart showed that the degree of disagreement about child-rearing between parents was significantly linked to children's maladjustment [36]. A possible mediating variable of parental disagreement in child-rearing on EBP is the poor conflict resolution strategies of parents. Child-rearing disagreement may frequently occur with the child present and may provoke negative emotions in the child [61]. Children may imitate the parents' argumentative, hostile, emotional conflict-resolution strategies, and such child behaviors could then be perceived as externalizing behavior problems by the parents. Alternatively, the child may withdraw in the face of disagreement between the parents

and become sad, which is then perceived by the parents as internalizing problems. No studies on the association between parental disagreement in child-rearing and child EBP has been conducted in Nepal, and further exploration of its effect on child mental health is warranted in future studies.

In the present study, physical punishment was found to associate with Externalizing Problems. Similar findings have been reported by several international studies [62]. A meta-analysis of a sample of 292 middle-class families with 8–12 years-old children from several countries (China, India, Italy, Kenya, Philippines and Thailand), found that physical punishment which included spanking, slapping, grabbing, shaking, and beating up, was associated with adverse child outcomes [38]. A study using nationally representative data in Nepal suggested that physical punishment of children is a common phenomenon with a prevalence ranging from 34% in the central hilly region to 60% in the mid-western hill region [63]. The prevalence is similar to other LMICs [39]. However, no previous Nepali studies have explored possible associations between physical punishment and child EBP. Hence, further research is paramount to explore such associations in more detail. The findings from this study suggest that protecting children from physical violence through national policies is warranted, focusing on eliminating corporal punishment in homes as well as in schools. Our recommendation is in line with a recent recommendation from the United Nation Children Fund (UNICEF) [64].

Globally, there are around 272 million international migrants with the largest proportion coming from Asia (41%) [65]. An estimated 3.5 million Nepali are working abroad, primarily in India, Malaysia and in the Middle East [65]. While many families have benefitted from the remittance that the migrants have earned, it has also resulted in family separation, affected child-rearing and fragmented the emotional and other support within the family [66]. In the present study, we found that migrant worker mother status was positively associated with child Externalizing Problems. However, no association was found for migrant worker fathers. A recent meta-analysis suggested that mental health and well-being of left behind children are not always negative and depend upon gender of the migrating parent, family norms, as well as other family characteristics [67]. While some South-East Asian countries showed negative effects of parental migration [68, 69], in other countries, left behind children were better off than non-left behind children [70, 71]. Perhaps, some unknown, resilience factors may have played a role, such as strong family- and social support mechanisms and increased family income which might have outweighed the negative impact.

## Strengths and limitations of the study

This is a large-scale study demonstrating the association of several family determinants of child EBP in Nepal. Due to a large number of children from across the country, the study generates solid information and is the first of its kind in Nepal. Moreover, the participation rate of the study was high (99.5%). The validity of the study was further strengthened by thorough procedures and the collection of data by trained research assistants.

However, there are some limitations to the study. As this was a cross-sectional study and not a longitudinal study, we could not provide insight into cause-and-effect relationships. Further, the design of our study did not allow us to examine in more details the different mechanisms of the associations between family factors and child EBP.

The small effect sizes in the present study indicate that in addition to the family variables accounted for, other environmental- and child variables might be important to add to the model. It would have been helpful if we had included other potential risk factors for children's EBP such as domestic violence, child neglect and substance abuse. The control variables (child age, child gender and major life events) included in this study are also sparse. More complete

set of covariates would have given more accurate results and more precise assessment of the included family correlates.

Also, if the family function/ dysfunction had been objectively measured, it could have resulted in a more predictive measure. However, it would be difficult to manage objective, high quality measurement of family function/ dysfunction in a large sample like this as it would require extremely large resources.

As parents' reports were the sole source of information for both background data and data on child EBP, it is likely that our data had some reporting bias. For instance, parental responses may reflect the extent to which parents are socially aware of acceptable family functioning and child- rearing. Use of complementary methods of obtaining information such as using teachers and the child himself/herself as informants, as well as using qualitative data from parents and children, would have enriched and strengthened the validity of the results.

The instruments that we used in this study were in Nepali language. Although everyone in Nepal speaks Nepali, the indigenous ethnic groups in our study have their own languages which differ from the Nepali language. Language and cultural barriers might have affected the understanding and conception of certain words and meanings for some of the participants. If so, we do not know in what way this might have affected the results. Some of the variables that were used as correlates might not have been precise enough to accurately measure the specific topics under investigation. For some variables, additional questions might have been posed to make more valid constructs. For instance, we lacked data on the onset, course, and outcome of parental illness. Limited research data in this area have resulted in inconclusive results regarding how patterns of parental illness are associated with child outcomes [30]. As for the family structure variable, adding questions on whether additional help was provided for single mothers by other family members, and questions about the additional presence of elderly family members in poor health requiring extra assistance, might have increased the validity of the variable. Finally, the prevalence for some of the dichotomous variables were low, which makes some of the variables less powerful in establishing associations. This might have affected some of the results.

## Clinical implications

This study provides new knowledge about environmental risk factors of child EBP in Nepal which may have implications for both clinical and preventive mental health work. In child psychiatric work, knowledge about possible risk factors in the child's family and social environment and how to integrate such knowledge in the assessment and treatment of the child, is paramount for problem solving. Hence, clinicians should routinely ask about family environment when trying to understand or assist children with internalizing or externalizing behavior problems. While genetic and biological factors may also be implicated in child EBP, current research, like the present study, establishes the importance of family factors. Hopefully, studies like ours might increase clinicians' awareness of family factors' influences on child EBP.

## Future research

Future empirical studies focusing on EBP in Nepal should investigate more into environmental risk and protective factors related to EBP. This will provide useful insights into risk factors and identify vulnerable groups of children, which are essential as a basis for the prevention of child mental health. As mentioned above, future studies should explore more detailed into the different risk factors such as family structure, migrant worker parents, parental illness, family conflict mechanisms, and parenting. Studies measuring family variables more objectively and thoroughly, possibly with more than one source of informants of family functioning, are

warranted. In addition, other environmental risk factors for child EBP not included in the present study, should be investigated. Further, future longitudinal studies on correlates of child EBP may provide insight into cause-and-effect relationships. Finally, inter-rater comparisons of child EBP are recommended due to the contextual variability in children's behavior. Till date, no such studies have been performed in Nepal.

## Conclusion

Similar to other countries, child EBP in Nepal is associated with several family risk factors. A positive correlation was found for factors such as parental education level, family structure, parental psychopathology, physical illness in parents, elevated level of conflicts in the family, parental disagreement in child-rearing, and harsh child-rearing control methods like physical punishment. Overall, the effect sizes were small in our study. Contrary to what was expected, we found no significant association between child EBP and migrant worker father status. Further studies are warranted to confirm some of the results.

## Supporting information

**S1 File.**
(SAV)

## Acknowledgments

We are grateful to all participating parents, and all the research assistants and field supervisors. Further, we would like to thank Dr. Arun Raj Kunwar and his child and adolescent psychiatry team at Kanti Children's Hospital, Kathmandu, for their support.

## Author Contributions

**Conceptualization:** Jasmine Ma, Anne Cecilie Javo.

**Formal analysis:** Jasmine Ma, Bjørn H. Handegård.

**Methodology:** Jasmine Ma, Anne Cecilie Javo.

**Supervision:** Pashupati Mahat, Per Håkan Brøndbo, Bjørn H. Handegård, Siv Kvernmo, Anne Cecilie Javo.

**Writing – original draft:** Jasmine Ma.

**Writing – review & editing:** Jasmine Ma, Pashupati Mahat, Per Håkan Brøndbo, Bjørn H. Handegård, Siv Kvernmo, Anne Cecilie Javo.

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
