## [Decision Letter · Decision Letter 0]

10 Nov 2021

PONE-D-21-27450FAMILY CORRELATES OF EMOTIONAL AND BEHAVIORAL PROBLEMS IN NEPALI SCHOOL CHILDRENPLOS ONE

Dear Dr. Ma,

Thank you for submitting your manuscript to PLOS ONE. After careful consideration, we feel that it has merit but does not fully meet PLOS ONE’s publication criteria as it currently stands. Therefore, we invite you to submit a revised version of the manuscript that addresses the points raised during the review process.

We look forward to receiving your revised manuscript.

Kind regards,

Pranil Man Singh Pradhan, M.D.

Academic Editor

PLOS ONE

Journal Requirements:

Reviewers' comments:

Reviewer's Responses to Questions

**Comments to the Author**

1. Is the manuscript technically sound, and do the data support the conclusions?

Reviewer #1: Yes

Reviewer #2: Yes

2. Has the statistical analysis been performed appropriately and rigorously? 

Reviewer #1: Yes

Reviewer #2: I Don't Know

3. Have the authors made all data underlying the findings in their manuscript fully available?

Reviewer #1: Yes

Reviewer #2: Yes

4. Is the manuscript presented in an intelligible fashion and written in standard English?

Reviewer #1: Yes

Reviewer #2: Yes

5. Review Comments to the Author

Reviewer #1: This is a very good paper to understand child behavior in the Nepalese context. However, I have certain methodological queries that requires the explanation from the authors before the paper could be published.

1. Page 15; line 168: You mention "The parents were given a small gift as an inducement to participate."

Is this process of inducing the participants acceptable? Have you discussed about this issue with the ethics committee or was it explained in your research proposal? It requires explanation to make the study ethically sound.

2. Page 15; line 172: You have given the abbreviation CBCL.

Have you explained this abbreviation before? It needs to be explained/ elaborated before introducing in the short-form.

3. Page 15; line 175: You mention Nepali version of the CBCL/6-18 without any reference.

What is the reference for the Nepali version CBCL? How, when and where was this version developed? I think it requires good explanation

4. Page 16; line 190:You mention "Cronbach’s alphas for the empirically based problem scales were........".

Was this Cronbach alpha value for the Nepali version? It is not clear. Has the Cronbach alfa value of the Nepali version you use been mentioned or explained? It requires explanation.

Reviewer #2: I would like to congratulate the authors for their good work. The research indeed was need for to understand the current child rearing scenario in Nepal. There are minor comments that I have provided. Kindly go through the attached pdf and check the comments.

6. PLOS authors have the option to publish the peer review history of their article (what does this mean?). If published, this will include your full peer review and any attached files.

Reviewer #1: **Yes: **Dr. Ajay Risal MD, PhD

Reviewer #2: No

---

## [Author Response · Author response to Decision Letter 0]

8 Dec 2021

Response to reviewers

Thank you both for your valuable suggestions as to revision of the manuscript. Below is our response. We have tried to give you our answers to the questions posed one by one and in doing so, we have referred to the particular lines in our revised manuscript with track changes where we have put the revisions (see Revised manuscript with track changes). 

Reviewer #1: 

Comment 1: Page 15; line 168: You mention: "The parents were given a small gift as an inducement to participate. Is this process of inducing the participants acceptable? Have you discussed about this issue with the ethics committee or was it explained in your research proposal? It requires explanation to make the study ethically sound”.

Our reply: 

Thank you for your comment. According to the Nepal Health Research Council (NHRC), Section 4.5: “Compensation and Payment”, researchers should make provision for compensating the participants’ efforts and time for research purposes. In the same section, it says that the information related to the provision for compensation should be communicated to the research participants. The compensation could be either monetary or non-monetary. However, it should be uniform to all participants. 

In our study, we gave a stationary, non-monetary, inexpensive gift (a notebook and pencils) to all the participants, meant as a compensation for any inconvenience in connection with their participation in the project. This gift was communicated and equally distributed to all participants. It was explained in our research proposal to the university (UiT- the Arctic University of Tromsoe, Norway) when applying for admittance for the first author (Dr. Ma) to study as a PhD candidate, and was also included in our application to the ethical committee of the NHRC, Nepal. 

We have now revised the sentence regarding gifts given to the participants as an acknowledgement of their participation (see line 170).

Comment 2: Page 15; line 172: You have given the abbreviation CBCL. Have you explained this abbreviation before? It needs to be explained/ elaborated before introducing in the short-form.

Our reply: 

Thank you for rightly pointing this out. We have now explained the abbreviation in the heading in the revised manuscript with track changes (line 178) and have used the word “instrument” instead of “CBCL” in line 175.

Comment 3: Page 15; line 175: You mention Nepali version of the CBCL/6-18 without any reference. What is the reference for the Nepali version CBCL? How, when and where was this version developed? I think it requires good explanation

Our reply: We have now added a sentence about the Nepali version of the CBCL with the reference in the revised manuscript (line 178-179).

The CBCL/6-18 (version 2001), TRF and YSR instruments were translated in Nepali in 2003 by Dr. Pashupati Mahat after he got a written consent from Dr. Thomas Achenbach to translate and validate these ASEBA tools in a Nepali school children population in connection with his PhD project (see reference number 43 in the revised manuscript). The permission from Dr. Achenbach is mentioned in the Nepali version of the tools at the end of each page as “Copyright T.M. Achenbach. Reproduced by permission of Nepali author P. Mahat”. The first author of the present paper got her permission from Dr. Mahat to use the CBCL and Dr. Mahat himself is one of the co-authors of her previous paper (Ma et al., 2021) and this present paper. 

Description of the translation process: All three tools (CBCL/6-18, TRF and YSR) were translated into Nepali by seven independent professionals (2 psychologists, 2 psychiatrist and 3 English linguistic professors). Five rounds of joint meetings were conducted to compare the translated document to make it more linguistic meaningful in a Nepali context. The translated tools were then back translated into English by three English language teachers (more than 15 years in English language teaching professors with extensive experiences in translation works in UN documents). The team first translated the CBCL/6-18, reviewed and back translated into English, then the same process followed for the TRF and the YSR. Dr. Pashupati Mahat (responsible researcher) reviewed all translations and edited the final versions of the three tools (TRF, YSR and CBCL). In agreement with the author of the ASEBA instruments, Dr. Achenbach, some final adjustments were made, and a copyright foot note was added in the final versions of Nepali translated tools.

Comment 4: Page 16; line 190: You mention "Cronbach’s alphas for the empirically based problem scales were........". Was this Cronbach alpha value for the Nepali version? It is not clear. Has the Cronbach alfa value of the Nepali version you use been mentioned or explained? It requires explanation.

Our reply: 

We have now clarified this point by explaining and formulating the sentence more precisely to avoid any misunderstandings – see revised manuscript with track changes (line 194-198). 

Reviewer #2: 

Comment 1: Line 149-150: It comes to be 15 in total but 16 is mentioned.

Our reply: 

As mentioned in the manuscript, our study was conducted in the three main geographical regions of Nepal where we purposively selected 3 districts from the mountain region, and 6 districts each from the Middle Hills and the Tarai region which add up to 15 districts. In addition, the Kathmandu district was included, which makes it 16 districts in total. We have made the sentence clearer in the revised manuscript with track changes (line 150-151). 

Comment 2: Line 244: Table 1 Family structure single parent 380 (9.9%). This is significantly in contrast with national data, where single motherhood was 25%. Any comments in the discussion section why the findings differed so much?

Our reply: Thank you for your comment. The 25% single motherhood from the national data presented in the Nepal Census, 2011, represents all single mothers and not just mothers with school-age children aged 6-16 years. Besides, the census represents earlier data. The latest census was published in 2011, i.e. 6 years before the collection of our data. Therefore, the national data may not be directly comparable with our study. In the discussion section, we decided to focus on the extended family structure and its correlation with child EBP as we found the strongest association with child EBP for this particular family structure. 

Comment 3: Line 267-268: I’m not an expert of statistics. But in my understanding the factors we select for Multiple regression analysis depends upon the p value seen in bivariate analysis. Once criteria allows variables with p value less than 0.025 to be selected for multivariate analysis. Here you have selected all the variables. Can you explain why?

Our reply: 

Thank you for your comment. Based on a literature search of what variables are likely to affect mental health in the Nepali context, we selected independent family variables from a larger set of variables. So, a selection process preceded the multiple regression. In the model, variables play different roles. Family variables are the main independent variables in this study, while child characteristics and demographic variables play a role as control variables. So, first and foremost, we wanted to assess the contribution of family variables over and above the effect of the control variables. So, control variables were forced into the model. For the family variables, we wanted to test the overall effect of all of these, adjusting for the control variables. Of course, we could have chosen a strategy of removing non-significant family variables so that only significant family variables remained in the final model, but instead we wanted to assess each selected variable, adjusted for the remaining variables. Since we have more than 3800 observations, overfitting issues are unlikely to be present.

Comment 4: Line 279: Table 3. Need to give full form

Our reply : 

Thank you for your suggestion. We have explained the different initials (F, B, SE) in footnotes of all the tables. See the revised manuscript with track changes (line 288 and line 294).

Comment 5: Line 299-300: This should also be in limitation section (i.e., about language difficulties for some parents belonging to ethnic minorities)

Our reply: 

Thank you for your suggestion. We have now added a few lines regarding the same in the limitation section - see revised manuscript with track changes (line 458-462). 

Comment 6: Line 303-305: I think you should keep it as the main reason as it is a valid explanation (i.e. about ratings of child problems by ethnic minorities)

Our reply: 

Thank you for your suggestion. We have now revised it accordingly - see revised manuscript with track changes (line 306-309).

Comment 7: Line 327-328: Contrary to finding in this study, children growing with grandparents have shown to develop wisdom, hence are able to be more stable. Review research by Dilip jeste on wisdom.

Our reply: 

Thank you for the information. Yes, there are studies where they have found results contrary to our findings. We have now revised and added the reference according to your advice (see line 341-343).

Comment 8: Line 333: It would be nice if you could add more explanations about low mental illness in the parents. What they perceive as mental illness? Could it be that for them Psychosis is only mental illness? How depression and anxiety are neglected and not seen as mental illness. How social stigma and lack of awareness about mental illness might have lead to low percentage observed. 

Our reply: 

Thank you for your suggestion. Yes, we agree that the way the parents perceive mental illnesses might have affected their ratings. According to your suggestions, we have now added a comment on this – see the revised manuscript with track changes (line 363-367). 

Comment 9: Line 342: Isn’t this the strongest point, as the latest research has been showing strong genetic link between mental disorders? It would be nice to keep this as the main point, at first. 

Our reply: 

Thank you for your suggestion. We have put it first as recommended – see the revised manuscript with track changes (line 348-353). The focus of our paper is the socio-environmental correlates of child EBP. However, we agree that the importance of the genetic link should be stressed, and we have done so in our revision, putting it first. 

Comment 10: Line 387: This is a very important finding, please elaborate your recommendations. How with this finding the current practice should be addressed. If possible keep it in a separate paragraph to highlight it.

Our reply: 

Thank you for your suggestion. We have now elaborated more on physical punishment in a Nepali context, including more specific recommendations – see the revised manuscript with track changes (line 412-416).

---

## [Decision Letter · Decision Letter 1]

3 Jan 2022

FAMILY CORRELATES OF EMOTIONAL AND BEHAVIORAL PROBLEMS IN NEPALI SCHOOL CHILDREN

PONE-D-21-27450R1

Dear Dr. Ma,

We’re pleased to inform you that your manuscript has been judged scientifically suitable for publication and will be formally accepted for publication once it meets all outstanding technical requirements.

Kind regards,

Pranil Man Singh Pradhan, M.D.

Academic Editor

PLOS ONE

Additional Editor Comments (optional):

Reviewers' comments:

Reviewer's Responses to Questions

**Comments to the Author**

1. If the authors have adequately addressed your comments raised in a previous round of review and you feel that this manuscript is now acceptable for publication, you may indicate that here to bypass the “Comments to the Author” section, enter your conflict of interest statement in the “Confidential to Editor” section, and submit your "Accept" recommendation.

Reviewer #1: All comments have been addressed

Reviewer #2: All comments have been addressed

2. Is the manuscript technically sound, and do the data support the conclusions?

Reviewer #1: Yes

Reviewer #2: Yes

3. Has the statistical analysis been performed appropriately and rigorously? 

Reviewer #1: Yes

Reviewer #2: I Don't Know

4. Have the authors made all data underlying the findings in their manuscript fully available?

Reviewer #1: Yes

Reviewer #2: Yes

5. Is the manuscript presented in an intelligible fashion and written in standard English?

Reviewer #1: Yes

Reviewer #2: Yes

6. Review Comments to the Author

Reviewer #1: The authors have adequately addressed my concerns so it may now be accepted for publication. Congratulations!

Reviewer #2: All the comments were adequately answered. I think there was some problem in the line numbers as the numbers were mentioned in around 10 count additional. Anyways, I found the answers.

7. PLOS authors have the option to publish the peer review history of their article (what does this mean?). If published, this will include your full peer review and any attached files.

Reviewer #1: **Yes: **Ajay Risal MD, PhD

Reviewer #2: No

---

## [Editor Report · Acceptance letter]

7 Jan 2022

PONE-D-21-27450R1 

FAMILY CORRELATES OF EMOTIONAL AND BEHAVIORAL PROBLEMS IN NEPALI SCHOOL CHILDREN 

Dear Dr. Ma:

I'm pleased to inform you that your manuscript has been deemed suitable for publication in PLOS ONE. Congratulations! Your manuscript is now with our production department. 

Kind regards, 

on behalf of

Dr. Pranil Man Singh Pradhan 

Academic Editor

PLOS ONE